# Estimating the Distribution of Japanese Encephalitis Vectors in Australia Using Ecological Niche Modelling

**DOI:** 10.3390/tropicalmed7120393

**Published:** 2022-11-22

**Authors:** Morgan Furlong, Andrew Adamu, Roslyn I. Hickson, Paul Horwood, Maryam Golchin, Andrew Hoskins, Tanya Russell

**Affiliations:** 1Australian Institute of Tropical Health and Medicine, James Cook University, Townsville, QLD 4811, Australia; 2College of Public Health, Medical and Veterinary Sciences, James Cook University, Townsville, QLD 4811, Australia; 3Commonwealth Scientific Industrial Research Organisation (CSIRO), Townsville, QLD 4811, Australia

**Keywords:** Japanese encephalitis virus, ecological niche modelling, species distribution modelling, *Culex*, *Culex annulirostris*, *Culex quinquefasciatus*, *Culex sitiens*, arbovirus, Australia

## Abstract

Recent Japanese encephalitis virus (JEV) outbreaks in southeastern Australia have sparked interest into epidemiological factors surrounding the virus’ novel emergence in this region. Here, the geographic distribution of mosquito species known to be competent JEV vectors in the country was estimated by combining known mosquito occurrences and ecological drivers of distribution to reveal insights into communities at highest risk of infectious disease transmission. Species distribution models predicted that *Culex annulirostris* and *Culex sitiens* presence was mostly likely along Australia’s eastern and northern coastline, while *Culex quinquefasciatus* presence was estimated to be most likely near inland regions of southern Australia as well as coastal regions of Western Australia. While *Culex annulirostris* is considered the dominant JEV vector in Australia, our ecological niche models emphasise the need for further entomological surveillance and JEV research within Australia.

## 1. Introduction

On 10 February 2022, two outbreaks of Japanese encephalitis virus (JEV) were confirmed in piggeries in New South Wales (NSW) which reported high incidences of stillbirths, abortions and piglets born with neurological disease [1]. Reports from the outbreak in Forbes, NSW state about a third of litters were impacted since at least the start of 2022 [1]. By 25 February, nine piggeries reported JEV outbreaks, this time including production sites in the states of Queensland (QLD) and Victoria (VIC). Over the next month, more than fifty piggeries across NSW, VIC, QLD and South Australia (SA) reported infections, in addition to twenty-four confirmed human cases and three deaths [2]. Prior to these reports, JEV had only been sporadically detected in areas of northern Australia, specifically the Torres Strait Islands, since the virus’ first detection in 1995. The four cases recorded on 2 March were the first reported human cases of JEV in southern Australia [3] and emphasise the virus’ expanding geographic range.

JEV is a mosquito-borne flavivirus that circulates enzootically between vectors and amplifying hosts including Ardeid waterbirds and both domestic and feral pigs [4]. More than ten mosquito species have been identified as competent vectors, but *Culex* species are considered to be the most important epidemiologically due to zoophilic feeding patterns [5]. While birds do not present with clinical disease, JEV infection in pigs can lead to abortion, weak and stillborn piglets, and infertility [6]. The virus can spill over to humans and other vertebrates via the bite of an infected female mosquito, but most are not capable of producing high enough levels of viremia to infect a new vector. Dead-end hosts include humans, cattle, horses and some species of marsupials [6]. Around 50,000 human cases are reported globally each year with symptoms ranging from non-specific flu-like illness to severe encephalitis [5,7]. It is estimated only 1% of human cases are symptomatic. Of those who do experience clinical symptoms, the associated fatality rate is between 20 and 30%, while 30 to 50% of survivors are left with long term, neurological disabilities [7,8]. No specific treatment is recommended; however, two vaccines are available for human use [5]. JEV is endemic in tropical regions, but viral spillover to humans in subtropical and temperate regions is associated with seasonality and above average rainfall, especially during La Nina weather events [5,9,10,11,12].

Associations between climatic variables, mosquito population dynamics and infectious disease outbreak risk have been demonstrated repeatedly [13,14,15,16]. Monthly, or even daily, climate variations can alter vector ecology and subsequently arbovirus epidemiology [12,17]. Aquatic survival rate, adult survival rate, susceptibility to viral infection, and the length of the extrinsic incubation period are significantly influenced by temperature for a plethora of mosquito species [14,18,19,20,21,22,23,24,25,26,27,28,29]. Additionally, rainfall and humidity have been noted to influence the longevity and abundance of vector breeding sites [9,13,30]. With regard to the *Culex* JEV vectors, higher temperatures have been associated with increased viral replication and transmission among *Cx. tritaeniorhynchus* populations in Asia [31]. For *Cx. annulirostris*, larval development was found to range from 37 days at 15 °C and 8.75 days at 35 °C [32]. Greater than average rainfall events were determined to be associated with higher *Culex* species abundance, and flooding is associated with mosquito population growth [23,25].

A widely accepted approach for mapping species distributions is ecological niche modelling (ENM) [31,33], where niche is defined as the spectrum of optimal environmental conditions ideal for population persistence [34]. For example, researchers were able to determine that the maximum temperature of the warmest month, the mean temperature of the coldest quarter, elevation, precipitation of the driest month and precipitation of the driest quarter had the greatest impact on *Cx. tritaeniorhynchus* distribution compared to other bioclimate variables using ecological niche modelling techniques [35]. Ecologists have theorised that highest levels of arboviral disease transmission are most likely when environmental conditions fall within the niche of competent vectors [34]. However, many mosquito species are considered “ecological generalists” and can tolerate a broad range of environmental conditions. Additionally, JEV vectors are capable of breeding in a diverse array of habitats including groundwater microhabitats, tidal marshes, roadside ditches, rice fields and man-made containers [28].

Australia’s diverse biome is made up of tropical, subtropical and temperate zones, and its landcover, dominated by forests, rainforests, and grasslands [23], supports over 300 different mosquito species with seventy-five currently recognised as important arboviral vectors [23]. In Asia and Pacific Island nations, *Culex* (*Cx.*) *tritaeniorhynchus* is considered the primary JEV vector, and *Cx. vishnui, Cx. gelidus, Cx. fuscocephala* and *Cx. pseudovishnui* have been established as secondary vectors [36,37]. A recently published review of important JEV vectors in Australia identified *Cx. annulirostris* as the primary JEV vector in the country after thorough consideration of viral competence, host-feeding patterns, species population dynamics and field detection studies [6]. *Cx quinquefasciatus, Cx. gelidus and Cx. sitiens* were highlighted as potential secondary vectors in Australia due to established competence and regional abundance [6,7]. There is, however, limited information regarding the distributional range of *Culex* species in Australia. This study aims to better inform our understanding of the distributional range of JEV competent vectors across Australia by combining known vector occurrence records with ecological drivers of distribution to derive continental scale, spatially continuous estimates of species distribution.

## 2. Materials and Methods

Models of species distribution were developed which linked species occurrence data with high resolution biotic and abiotic environmental data [34]. Mosquito presence data for *Cx. annulirostris, Cx. quinquefasciatus* and *Cx. sitiens* were taken from the online databases VectorMap and the Atlas of Living Australia’s (ALA) [38,39,40,41,42,43,44,45,46,47,48,49,50,51,52,53,54]. These databases did not contain presence data for *Cx. gelidus*, as such, this species was excluded from further analyses. Records from ALA were downloaded using quality flags which automatically excluded records identified as unsuitable for use in species distribution modelling. This included records with spatial position quality issues and duplicate records. Presence records from VectorMap were downloaded via the MosquitoMap attributes table. Responses were filtered to species = “annulirotris”, “quinquefasciatus” or “sitiens” and country = Australia. Coordinates for each species were combined from the two databases and compiled into separate .csv files formatted as species, longitude, and latitude for each species to be compatible with the species distribution modelling software requirements. Nine second resolution (9s) climate variables with radiative and elevation lapse rate correction for continental Australia were obtained from the Commonwealth Scientific Industrial Research Organisation (CSIRO) Data Access Portal [55]. Climate variables were based on Xu and Hutchinson’s ANUCLIM 6.1 thirty-year average between 1 January 1976 to 31 December 2005. All climate maps span between 8°0′0″ S to 43°44′33″ S latitude and 112°54′0″ E to 154°0′0″ E longitude. Temperature, precipitation and evaporation are given as annual means or totals and maximum and minimum monthly values (Appendix A). Highly correlated climate variables were removed via analysis in Python, where highly correlated was defined as greater than 90% correlation, reducing the included variables in each Maxent run from 25 to 15.

Species distributions were estimated using the java-based, machine learning, maximum entropy modelling software, Maxent, version 3.4.4. Maxent runs using a presence-background algorithm which uses simulated absence data when raw absence data is limited [34]. Maxent removed any duplicate presence records [56]. Model training used deviance, goodness of fit and generalised linear models to assess performance, involved the use of 10,000 randomly sampled background points and 75% of occurrence records. The remaining 25% of *Culex* species presence data was used solely for model testing. Species distribution models were created using the complementary log-log (cloglog) output, which provides a strong prediction of presence probability based on raw values, cumulative thresholds, inhomogeneous Poisson process and independent species occurrence points [57]. Although mosquitoes naturally congregate in areas ideal for laying eggs and feeding [58], the cloglog transformation was chosen over the logistic transformation option in Maxent since each presence coordinate was provided as an independent record [57]. The choice of feature classes was catered to mosquito-specific biology as suggested by Merow et al. [59]. While there is evidence mosquitoes benefit from high temperatures, too high of temperatures have been noted to be associated with increased mortality [35]. Similarly, while abundant rainfall is essential for reproduction, high levels of rainfall may washout larvae leading to a decrease in mosquito abundance [60]. Therefore, Maxent was given the option of choosing between linear and quadratic feature classes since exact temperature and precipitation values associated with the above scenarios are not well defined. Maxent chose the linear feature type for each simulation. Additionally, Maxent was run with different regularisation multiplier values (0.1, 1, 10 and 100 where 1 is the default value). Models were compared based on area under the curve (AUC) values where a higher AUC value indicates better performance [33]. The highest AUC for both model testing and training were obtained from simulations where the regularisation multiplier was set at 0.1. Additional models were simulated using Maxent’s k-fold cross-validation technique for further ENM comparison and the creation of uncertainty maps for each species. The number of replications was set to 10, and both cross-validation ENMs and uncertainty maps are provided in the supplementary materials (Appendix A).

The sampling efforts used in this study are likely influenced by bias, and therefore, Maxent results cannot be interpreted for insights on species abundance. However, A bias file, obtained from Weiss et al. (2018), was added to each Maxent run with the aim of accounting for sampling and spatial bias [61]. The bias file represents the time it takes to travel to a population centre from any given location. The raster file was clipped to the extent of the Australian continent and then imported into ArcGIS to ensure its reference coordinate system, resolution and extent matched the climate files. The reference coordinate system, resolution and extent were adjusted using the project, resample and clip tools in the ArcGIS Toolbox, respectively. Travel time values were inverted so areas closer to urban centres contributed more towards bias using the Map Algebra raster calculator tool in ArcMap. Because Maxent can only process .asc files with positive and non-zero values, the raster calculator was used once more to increase all file values by one, and the conversion tool was used to convert the file from .tif to .asc.

The probability of species presence for each of the three vectors of interest was estimated between 0 and 1 for each pixel on the grid of species distribution maps. Raster files obtained via Maxent output were imported into ArcGIS version 10.7 to edit map symbology. Australian state borders were added to model raster files by accessing ArcGIS online via ArcMap. Specific geographic locations are referred to by their colonial names and not by their traditional Indigenous names.

Additional Maxent outputs of interest included jackknife analysis and sample averages files for each vector species. Only the outputs of Maxent runs using 25% of data for testing were evaluated; cross-validation jackknife and sample averages are not included in this review since an average file is produced for each model individually but not for the final output. Jackknife analysis assessed the importance of each climate variable in determining species distribution by measuring model performance using each climate variable in isolation and when each variable was omitted. Jackknife analysis plots using the receiving operating characteristic AUC for test data were chosen for further analysis. Jackknife analysis must be interpreted carefully for correlated environmental variables [39]. It is also important to note that environmental variable contribution rank is based on the most efficient algorithm for Maxent to run, therefore producing heuristic estimates. Sample averages files contained the average values for each environmental variable across all pixels based where species occurrence was recorded.

## 3. Results

### 3.1. Ecological Niche Models

The ecological niche model (ENM) for *Cx. annulirostris* was simulated using 297 presence records for model training, 99 presence records for model testing and 10,297 random background points (Figure 1a). Regions with the highest probability of *Cx. annulirostris* presence were estimated to stretch from the Torres Strait Islands north of Cape York, Queensland (QLD) to southern New South Wales (NSW) along the coast, regions surrounding Darwin in the Northern Territory (NT), and fragments of the northwestern coast of Western Australia (WA). Furthermore, a high probability of *Cx. annulirostris* presence was estimated inland where the borders of QLD, NSW and SA meet and regions of eastern Victoria (VIC) (Figure 1a).

The ENM for *Cx. quinquefasciatus* was simulated using 190 presence records for training, 63 presence records for testing and 10,190 background points (Figure 1b). Geographic areas estimated to have the highest probability of *Cx. quinquefasciatus* presence include coastal regions spanning Australia’s eastern coast from Cooktown, QLD to Melbourne, VIC, coastal and inland regions surrounding the Yorke, Eyre and Fleurieu Peninsulas, as well as Kangaroo, Flinders and King Islands, in SA and Tasmania (TAS). Coastal areas in between Esperance and Perth, WA were also identified as high likelihood of *Cx. quinquefasciatus* occurrence as well as inland regions near the VIC-NSW border.

*Cx. sitiens* ENMs were simulated using 72 and 23 presence records for training and testing, respectively in addition to 10,072 random background points (Figure 1c). Darwin, NT, the Torres Strait Islands and coastal regions in between Cooktown, QLD to south of Sydney, NSW were estimated to have a high probability of Cx. sitiens presence. No regions within SA, WA, TAS, Australian Captial Territory (ACT) or VIC were estimated to have greater than a low probability of presence for Cx. sitiens (Figure 1c). Common regions among all three vector species that were estimated to have a very low probability of presence include inland regions of TAS and Australia’s desert regions: Tanami, Great Sandy, Gibson, Great Victoria and Simpson Deserts (Figure 1). With regard to the uncertainty maps produced via the cross-validation Maxent runs, uncertainty remained low with low values of standard deviation among all simulated models for all species (Appendix A). For *Cx. quinquefasciatus*, portions of the region known as the Murray-Darling Basin were highlighted with low levels of uncertainty (Appendix A). Both *Cx. annulirostris* and *Cx. quinquefasciatus* uncertainty maps presented with low levels of uncertainty surrounding salt lakes in SA. The highest ENM uncertainty was found among *Cx. sitiens* outputs along the western coast of the Cape York Peninsula, QLD.

### 3.2. Contribution of Environmental Variables

Jackknife analysis revealed that aridity index and water deficit were important climate variables for predicting the distribution of all three mosquito species (Figure 2a–c). Environmental variables containing information on monthly temperature ranges were also found to be important in predicting species distribution ranges for *Cx. annulirostris* and *Cx. quinquefasciatus* (Figure 2a,b). *Culex sitiens* presence estimations relied heavily on information from the annual total actual evapotranspiration (modelled using terrain-scaled MODIS) layer (Figure 2c). The jackknife test of variable importance also showed temperature variables, including annual temperature range and maximum monthly maximum temperature, were useful in determining the geographic distributions for *Cx. quinquefasciatus* (Figure 2b).

Sample average values are included in Appendix A. The average mean annual aridity indices for *Cx. annulirostris*, *Cx. quinquefasciatus* and *Cx. sitiens* were 0.77, 0.73 and 0.97, respectively (Appendix A). The average minimum monthly and maximum monthly aridity indices based on all presence points were lower for *Cx. annulirostris* (0.20, 2.15, respectively) than *Cx. quinquefasciatus* (0.24, 2.66) and *Cx. sitiens* (0.28, 2.47) (Appendix A). It should be noted that average values for annual atmospheric water deficit were the lowest among *Cx. quinquefasciatus* occurrence points (Appendix A). The average for annual total actual evapotranspiration using MODIS was highest for *Cx. sitiens* (Appendix A). When comparing *Cx. annulirostris* and *Cx. quinquefasciatus*, total actual evapotranspiration, annual potential evaporation and the minimum monthly potential evaporation averages were higher for *Cx. annulirostris* (Appendix A). Average values for minimum monthly mean diurnal temperature were similar for *Cx. annulirostris* (9.31 °C) and *Cx. quinquefasciatus* (9.29 °C) but lower for *Cx. sitiens* (7.98) (Appendix A). With regard to the average temperature range determined based on mosquito species occurrence data, the annual temperature range was similar for *Cx. quinquefasciatus* (22.72 °C) and *Cx. annulirostris* (22.17 °C) followed by *Cx. sitiens* (18.86 °C) (Appendix A).

## 4. Discussion

Our ecological niche models (ENMs) revealed that all three *Culex* vectors considered in this report were estimated to have a high probability of presence (>0.6) in both densely populated urban areas and rural areas along Australia’s eastern coast (Appendix A). Figure 1 provided a detailed view of the probability of presence using the full continuous information from the ENM. For ease of interpretation, maps with manual probability classifications of very low (<0.09), low (0.09–0.29), medium (0.30–0.6) and high (>0.6) probability are provided in Appendix A. In the southern and western regions, *Cx. quinquefasciatus* populations were estimated to have a high probability of presence, suggesting these populations can occupy a unique niche. The majority of confirmed human JEV cases connected to the 2022 outbreak were tied to the Riverina region—the area of land between the Murrumbidgee River to the north and Murray River to the south [2]. The niche models estimate this region to be medium to high probability for both *Cx. annulirostris* and *Cx. quinquefasciatus* presence and low probability for *Cx. sitiens* (Appendix A).

Based on our models, both *Cx. annulirostris* and *Cx. quinquefasciatus* were estimated to have some likelihood of presence in over one-third of the Australian continent supporting the idea that various landcover types and water habitats support mosquito populations in Australia. The typical aquatic habitat utilised by *Cx. annulirostris* is shallow, vegetated freshwater, although larvae have been detected in a wide range of freshwater habitats including irrigation drains, dams, semi-permanent and permanent groundwater pools and flooded grasslands [15,21,24]. *Culex quinquefasciatus* immatures are commonly found among water with high concentrations of organic content such as sewers, ground pools, ditches or cesspools [6,62] including those contaminated with pig feces. *Culex* species’ ability to feed on a variety of vertebrate hosts, especially *Cx. sitiens, Cx. annulirostris* and *Cx. quinquefasciatus*, may also contribute to their widespread presence and the greater potential for encountering arboviruses with zoonotic risk [6,63]. *Culex sitiens* were estimated to have a narrower geographic distribution along Australia’s eastern coast and a medium to high probability of presence in coastal regions surrounding Darwin, NT. Populations of *Cx. sitiens* are typically associated with coastal habitats ranging from flooded saltmarshes, puddles or rock pools [62].

Jackknife analysis revealed aridity and water deficit, specifically the mean annual aridity index and annual atmospheric water deficit, ranked among the most important variables for the creation of all three species models. Variables containing information regarding evaporation and temperature were also identified as holding important information critical for species distribution model development. Minimum average temperature values produced by Maxent were the lowest for *Cx. quinquefasciatus* which may suggest increased likelihood of presence in temperate regions compared to *Cx. sitiens* and *Cx. annulirostris.* Alternatively, ENMs focused on *Cx. quinquefasciatus* in China estimated populations were most likely to be discovered inland or along the southeastern coast closer to the tropics [11]. It appears the ratio between evaporation and precipitation was more informative to the models than precipitation levels alone. While all three species’ distributions were strongly influenced by humidity (aridity index > 0.6), differences in evaporation, precipitation and temperature, these variables are correlated via the hydrological cycle, and therefore, conclusions regarding the role of moisture in predicting each species’ niche are limited. Previous research has established that *Culex* populations thrive in the presence of widespread, flooded habitats caused by above average rainfall, highlighting the connection between higher than average La Nina rainfall in Australia in 2022 and subsequent rise in JEV cases [24].

Arbovirus disease modelling is complicated by complex transmission cycles but strengthened with a robust understanding of disease ecology. JEV outbreaks are difficult to predict due to complex interactions between environmental conditions, vectors, hosts and anthropogenic activity related to political, economic and social determinants. While our ENMs reveal geographic locations where three JEV vectors are likely to be present, they do not give insight into JEV risk or mosquito population abundance. In Australia, *Cx. annulirostris* presence and abundance vary within and between years [25]. Larvae are found during the hotter and wetter months of the year, typically between January to March [64]. Adults are generally present throughout the year and are reported to survive overwinter, suggesting the potential for JEV transmission year-round, although *Cx. annulirostris* population growth slows over the winter months and dry season [25]. A study looking at differences in *Cx. annulirostris* survival between populations in Townsville (QLD), Brisbane (QLD) and Murray Valley (VIC) concluded that each population was dependent on different thermal requirements [15]. Furthermore, some *Culex* species are capable of dispersing long distances from their larval habitats. Both low and high wind speeds have been associated with mosquito dispersal up to hundreds of kilometres, especially winds associated with tropical cyclones [23]. One study noted that with wind, *Cx. annulirostris* mosquitoes dispersed an average of 150 km from their larval sites with the maximum distance dispersed being 594 km [6]. On the other hand, the flight range of *Cx. quinquefasciatus* is estimated to be less than 2 km [62]. Some arbovirus experts suspect wind as the culprit behind the introduction of JEV from Papua New Guinea to the Torres Strait Islands in 1995 [23]. The ENMs simulated for JEV vectors in Australia should be compared with reported human cases and populations of amplifying hosts to consider the full ecology of JEV transmission.

It is critical to mention that species distributions created by Maxent are strictly estimates since no absence data was provided during simulations. An additional limitation is that Maxent does not consider vector-specific population dynamics including survivorship, fecundity nor predation rate, and these models may fail to capture local-level mosquito dynamics such as life history traits and mosquito immunity. While model sensitivity was high for all ENMs, model uncertainty could be further reduced by combining information from mechanistic species distribution models and further entomological research focused on *Culex* species. While these models do not predict JEV risk, they can inform vector control and sampling strategies. Surveillance efforts should especially be strengthened among areas noted as uncertain since the western coast of the Cape York Peninsula is notorious for other arbovirus outbreaks including New Mapoon Virus and the West Nile Virus strain Kunjin as well as inland regions of the Murray-Darling Basin where the incidence of JEV cases was the highest (Appendix A). While attempts were made to account for spatial and sampling bias, result interpretation is further limited by a lack of true absence data and the inclusion of solely Australian occurrence records. Since the occurrence data used in this review is restricted to the Australian continent, interpretations are limited to the study area, and thus, exclude each species’ globally occupied niche potentially leading to an underestimation of their distributions (Appendix A). However, arbovirus and entomological literature have emphasized the strength of regional occurrence and climate data for estimating distributions, as different strains of mosquito species are noted to have different environmental tolerances within Australia and throughout the Pacific Islands [6,15,18,19,23]. More research on temperature-dependent traits and the role of microclimates is needed for each JEV vector species to determine their unique ecological niche. Other suspected JEV vectors in Australia include *Cx. gelidus*, *Cx. tritaeniorhynchus*, *Cx. bitaeniorhynchus*, *Cx. australicus*, *Cx. molestus, Aedes vigilax*, *Aedes notoscriptus* and *Mansonia* species [6].

Future studies should investigate associations between genomic differences and JEV transmission ability among different mosquito species and JEV genotypes. Evidence suggests *Cx. annulirostris* is a more competent vector for JEV G-II compared to JEV G-I [6]. There are five different JEV genotypes, and G-IV was recorded as the circulating genotype during the southern Australian outbreak. JEV competency was also found to vary among different lineages of *Cx. annulirostris*, with lineages ann-PNG1 and ann-PNG2 discovered to be competent for JEV G-I but not for G-II [6]. Future research should also aim to understand how rates of climate and land use changes, species adaptation and resilience will influence pathogen, vector and host geographic distribution and burden of disease. The Representative Concentration Pathway (RCP) scenario level 8.5, defined as 940 ppm of CO_2_ in the atmosphere by 2100, predicts a mean temperature increase of 2.8 to 5.1 °C, both excessively high and low levels of precipitation, and lower humidity levels, specifically in inland Australia, throughout the winter and spring. There is no doubt climate change will impact mosquito population abundance, diversity and peak and duration of JEV outbreaks.

## 5. Conclusions

We have estimated the presence probability of three of the likely JEV vectors in Australia. Although our study has several limitations, it also helps identify critical data gaps for understanding JEV transmission in Australia. Our results emphasise the importance of regional entomological surveillance, citizen science and a One Health perspective for unravelling ecological dynamics of zoonotic arboviruses.

## Figures and Tables

**Figure 1 tropicalmed-07-00393-f001:**
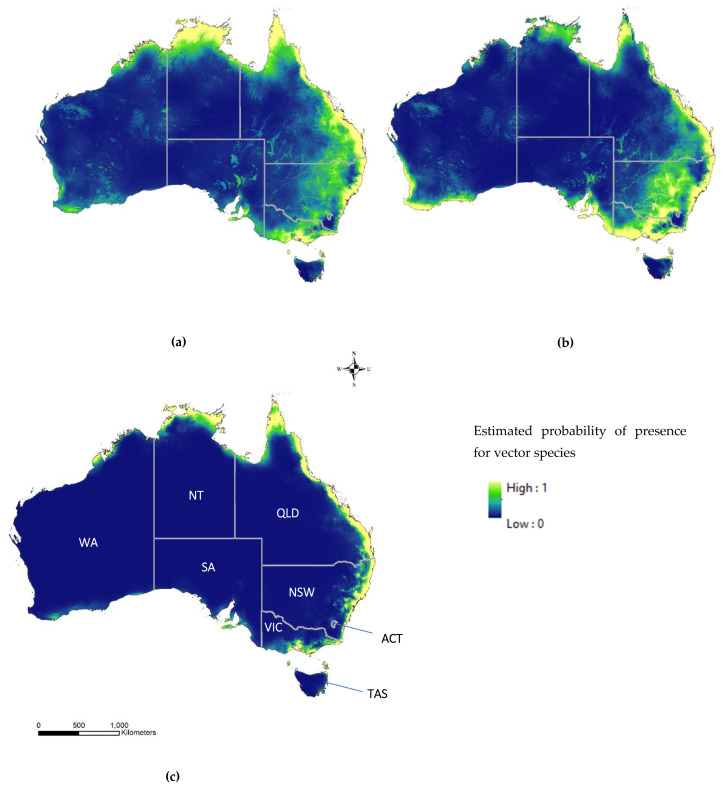
Ecological niche models for (**a**) *Cx. annulirostris*, (**b**) *Cx. quinquefasciatus* and (**c**) *Cx. sitiens* estimated using the cloglog transformation in Maxent. Colours represent the continuous probability of mosquito presence where blue is very low and yellow is high. States are labelled by their abbreviated names in figure (**c**) were NT = Northern Territory, QLD = Queensland, NSW = New South Wales, ACT = Australian Capital Territory, VIC = Victoria, TAS = Tasmania, SA = South Australia and WA = Western Australia.

**Figure 2 tropicalmed-07-00393-f002:**
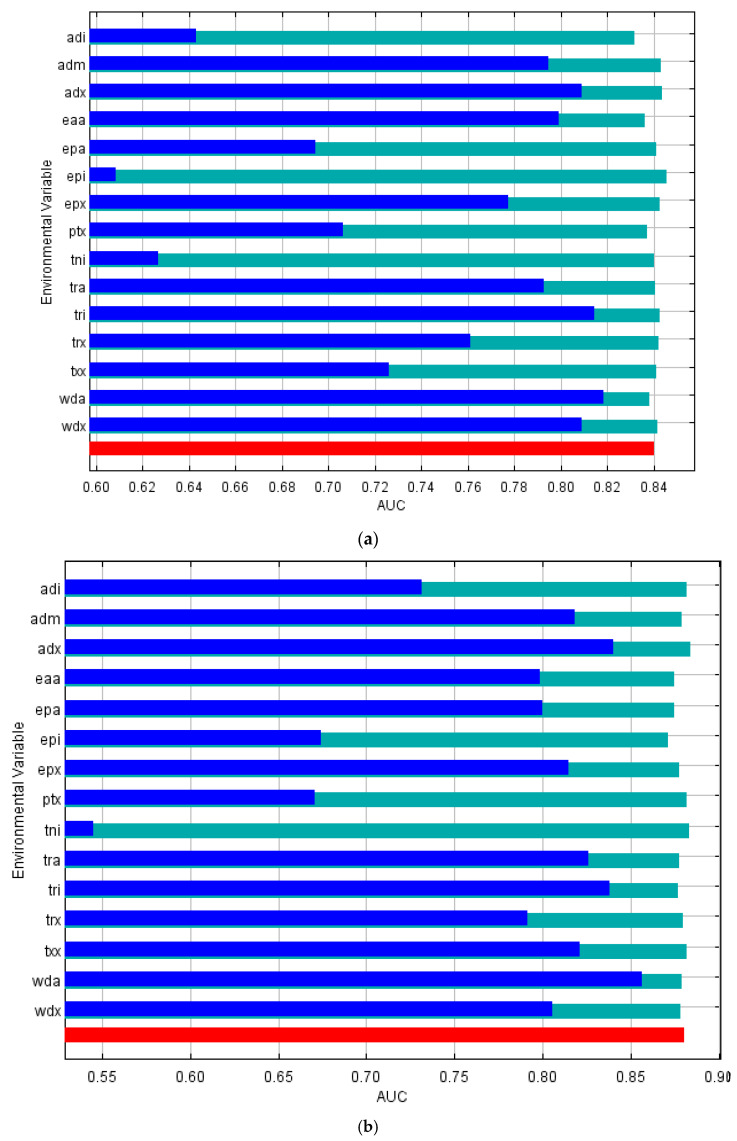
Jackknife analysis results for (**a**) *Culex annulirostris,* (**b**) *Culex quinquefasciatus* and (**c**) *Culex sitiens* where the blue line represents species distribution model gain when each climate layer is included on its own. Dark blue lines represent model performance when only the climate variable of interest was included, where the longer the dark blue bar, the higher the model gain which suggests the variable contains information critical to mapping that species’ distribution. The red bar represents maximum model performance, while the light green bar represents model performance when that climate variable was excluded. Acronyms for each climate layer are defined as follows: adi (minimum monthly aridity index), adm (mean annual aridity index), adx (maximum monthly aridity index), eaa (annual total actual evapotranspiration, terrain-scaled using MODIS), epa (annual potential evaporation), epi (minimum monthly potential evaporation), epx (maximum monthly potential evaporation), ptx (maximum monthly precipitation), tni (minimum temperature—monthly minimum), tra (annual temperature range), tri (minimum monthly mean diurnal temperature), trx (maximum monthly mean diurnal temperature), txx (maximum temperature—monthly maximum), wda (annual atmospheric water deficit) and wdx (maximum monthly atmospheric water deficit).

## Data Availability

The data presented in this study are available on request from the corresponding author.

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
