# Peer review of "Estimating the Distribution of Japanese Encephalitis Vectors in Australia Using Ecological Niche Modelling"

_tropicalmed, 2022, doi:10.3390/tropicalmed7120393_

Round 1

Reviewer 1 Report

The proposal is of interest, but major methodological issues should be accounted for before consideration for publication. The authors have performed the modelling procedures almost in an automatic manner, without considering specific theorical concepts and a number of publications advocating for the contrary. For instance:

- The occurrence data is restricted to the records occurring in the area of interest, thus not considering the entire niche occupied by the species at a global level, and probably underestimating the resulting distribution (see Parasites & Vectors 2020 9: 573, and Transbound Emerg Dis 2022 69: e1113-e1129). This resulting distribution based on local occurrences might accurately predict the (partial) niche currently occupied by the species in the area of interest. However, most probably, it underestimates the potential distribution, as parts of the species niche occupied elsewhere have not been included in the modelling procedures.

 - In the present study, sampling bias is not accounted for. Sampling bias and spatial autocorrelation in environmental information, and the consequent non-independence of occurrence data, could bias the resulting predictions. When sampling is biased, one cannot differentiate whether species are observed in particular environments because those locations are preferable or because they receive the largest search effort (see Ecography 2013 36: 1058–1069, and Ecography 2015 38(5): 541-545).

- The authors did not define an adequate accessible area (M) for each species included in the study. Following the BAM diagram (Soberón and Peterson Biodivers Inform 2005 2: 1-10), M represents the parts of the world that have been accessible to the species via dispersal over relevant periods of time. The former is considered crucial to minimize the impact of assumptions about absences from areas that are not accessible to the species (Ecol Modell 2011 222: 1810–1819).

- The modelling procedures have been performed with default settings, with no fine-tuning of model complexity. Maxent allows users to fit models of varying complexity, by specifying a number of feature classes and/or levels of a regularization coefficient. Default regularization often retains hundreds of correlated features, so when biological interpretation is important, it may be more helpful to seek simpler models (Ecography 2013 36: 1058–1069). Furthermore, it has been demonstrated that models that are inappropriately complex or inappropriately simple show reduced ability to infer habitat quality, reduced ability to infer the relative importance of variables in constraining species' distributions, and reduced transferability to other time periods (Ecol Appl 2011, 21: 335-342). Thus, a fine model calibration with different combinations of feature classes and values of Maxent’s regularization multiplier is strongly recommended (Ecol Appl 2011, 21: 335-342, Ecography 2013 36: 1058–1069).

In all, these methodological issues warns against the validity and reliability of the results.

Author Response

Reviewer 1

Comment 1 - The occurrence data is restricted to the records occurring in the area of interest, thus not considering the entire niche occupied by the species at a global level, and probably underestimating the resulting distribution (see Parasites & Vectors 2020 9: 573, and Transbound Emerg Dis 2022 69: e1113-e1129). This resulting distribution based on local occurrences might accurately predict the (partial) niche currently occupied by the species in the area of interest. However, most probably, it underestimates the potential distribution, as parts of the species niche occupied elsewhere have not been included in the modelling procedures.

Response - To investigate this claim, we downloaded occurrence records for each of the three mosquito species included in our study from the Global Biodiversity Information Facility and environmental climate data from WorldClim to assess the estimated niche at a global level. To analyze differences between the niche represented in our study and each species' global niche, we plotted the density of occurrence points noted for various temperature levels. (These graphs are now included in the supplementary materials.) Culex (Cx.) annulirostris and Cx. sitiens populations in Australia appear to align within the suitable temperature ranges found amongst other population of these species' distributions worldwide. However, it appears as though our data predicts Cx. quinquefasciatus to exist among lower suitable temperature ranges than trends noted among global populations. We acknowledge these findings may underestimate the later species' niche; however, we use regional and continental data rather than global data to estimate species distributions based on evidence noted in Rae 1990, Russell 1987, Socket et al. 2018, van den Hurk et al. 2022 and Madzokere et al. 2020, who have noted significant variation within populations of Culex species within Australia and throughout the Pacific Islands (references included in final manuscript). As emphasized in our discussion, we support future research looking into the role of microclimates to reveal species' unique niche. We have added a sentence addressing the limitation of using only

records occurring in the area of interest in the discussion section of the manuscript.

Comment 2 - In the present study, sampling bias is not accounted for. Sampling bias and spatial autocorrelation in environmental information, and the consequent non-independence of occurrence data, could bias the resulting predictions. When sampling is biased, one cannot differentiate whether species are observed in particular environments because those locations are preferable or because they receive the largest search effort (see Ecography 2013 36: 1058–1069, and Ecography 2015 38(5): 541-545).

Response - We strongly agree with reviewer 1's feedback to explicitly account for bias. To address this point, we have now included a bias file for all maxent simulations. The bias file was obtained from Weiss et al. (2018) and represents the travel time to a population centre. CSIRO had a version of the accessibility map clipped to the extent of Australia, and after aligning the raster with the climate files, inversing accessibility measurements and converting the file type to .asc, we were better able to account for bias in our review. We have updated our methods section and discussion section where bias is discussed as a limitation.

Comment 3 - The authors did not define an adequate accessible area (M) for each species included in the study. Following the BAM diagram (Soberón and Peterson Biodivers Inform 2005 2: 1-10), M represents the parts of the world that have been accessible to the species via dispersal over relevant periods of time. The former is considered crucial to minimize the impact of assumptions about absences from areas that are not accessible to the species (Ecol Modell 2011 222: 1810–1819).

Response - Due to the lack of presence and true absence data for Culex species in Australia, the adequate accessible area (M) for each species was based on species distribution maps produced in Webb et al. (2016) Guide to Mosquitoes of Australia. The estimated distribution for both Cx. annulirostris and Cx. quinquefasciatus stretched across the entire continent of Australia. While Cx. sitiens is primarily associated with estuarine wetlands, Maxent estimated their niche to be within these coastal areas, and it is interesting to note Cx. sitiens models had the highest AUC values out of all three species even without defining M. One may argue that inaccessible areas for vector species would include mountain ranges above 1,500 m; however, there is increasing evidence of occurrence records of Culex mosquitoes at high altitudes due to expanding their niches into cooler regions to escape climate change-associated warming. Elevation was also accounted for in our climate variables. From an epidemiological perspective, considering all possible locations within the study area of interest prevents vulnerable host populations from being considered low risk for Japanese encephalitis virus and other arbovirus transmission when it is unestablished if mosquitoes are present in the area or not. Inland surveillance must become more established before the area M can be included in Culex niche models, specifically for Australia.

Comment 4 - The modelling procedures have been performed with default settings, with no fine-tuning of model complexity. Maxent allows users to fit models of varying complexity, by specifying a number of feature classes and/or levels of a regularization coefficient. Default regularization often retains hundreds of correlated features, so when biological interpretation is important, it may be more helpful to seek simpler models (Ecography 2013 36: 1058–1069). Furthermore, it has been demonstrated that models that are inappropriately complex or inappropriately simple show reduced ability to infer habitat quality, reduced ability to infer the relative importance of variables in constraining species' distributions, and reduced transferability to other time periods (Ecol Appl 2011, 21: 335-342). Thus, a fine model calibration with different combinations of feature classes and values of Maxent’s regularization multiplier is strongly recommended (Ecol Appl 2011, 21: 335-342, Ecography 2013 36: 1058–1069).

Response ­4 -

To strengthen the robustness of our review, we began by finding highly correlated environmental variables, where highly correlated was defined as greater than 90%. This reduced the included environmental variables in Maxent simulations from 25 to 15. The choice of feature classes used in recent Maxent runs was chosen based on specific mosquito biology as suggested by Merow et al. (2013). While there is evidence mosquitoes benefit from high temperatures, too high of temperatures have been noted to be associated with increased mortality (Liu et al. 2018). Similarly, while abundant rainfall is essential for reproduction, high levels of rainfall may washout larvae leading to a decrease in mosquito abundance (Kurane et al. 2013). We decided to allow maxent to choose between the linear and quadratic feature classes since exact temperature and precipitation values associated with the above scenarios are not well defined, especially for Culex species. Maxent chose the linear feature type for each simulation.

Once all runs were successful with the added bias file and correlated environmental variables removed, we ran maxent for each species with a regularization multiplier of 0.1, 10 or 100 and compared AUC values to the default run where the regularization multiplier is set at 1. Liu et al. (2020) used a similar approach to evaluate model comparison where a higher AUC value indicates better performance. The highest AUC for both model testing and training were obtained from simulations where the regularisation multiplier was set at 0.1. These are the niche models included in our updated submission. We have updated the method and results sections pertaining to chosen maxent settings and contribution of environmental variables. We would like to thank you for your thorough examination of our manuscript and for providing key readings relevant to species distribution modelling as well as your expertise.

Reviewer 2 Report

The paper was well written and presents predicted JVE vectors distribution for a country with limited information regarding the distributional range of Culex species in Australia.

Author Response

Comment - The paper was well written and presents predicted JVE vectors distribution for a country with limited information regarding the distributional range of Culex species in Australia.

Response - Thank you for thoroughly reviewing our manuscript and providing your expertise.

Reviewer 3 Report

Only recently, severe Japanese encephalitis virus (JEV) outbreaks happened in Australia. Authors here demonstrated that ecological niche modelling can be used for describing the distribution of the most important vectors of JEV (three Culex species) in Australia. Even when mosquitoes can tolerate a broad range of environmental conditions, the results clearly show distinct geographic locations for the three Culex species. The presented results will certainly help in the development of novel vector control strategies. 

I suggest only minor revisions for the manuscript:

(1)  Cancel "Culex" in the keywords. It is enough to mention the three Culex species

(2)  Chapter 3.2: use the same spelling for the figures as before (without brackets)

(3) Lines 303 and 407: say Figure S1 and Figure S2

(4) References must be carefully checked for a uniform style of writing (titles of papers in lowercase letters, species names in italics)

Author Response

Reviewer 3

Comment 1 - Cancel "Culex" in the keywords. It is enough to mention the three Culex species.

Response 1 - We have decided to keep "Culex" as a key word in addition to the specific mention of each species since JEV vector competence is an evolving field, and we would like researchers looking into other Culex species to compare their findings.  

Comment 2 - Chapter 3.2: use the same spelling for the figures as before (without brackets).

Response 2 - We have updated the formatting for referencing figures in this paragraph to match chapter 3.1.

Comment 3 - Lines 303 and 407: say Figure S1 and Figure S2.

Response 3 - We have updated the noted lines above to S1 and S2 rather than "Supplementary Figure S1".

Comment 4 - References must be carefully checked for a uniform style of writing (titles of papers in lowercase letters, species names in italics)

Response 4 - We have now updated the formatting of titles included in the reference section. Thank you for thoroughly reviewing our manuscript and providing your expertise.

Round 2

Reviewer 1 Report

I'm glad most of mine suggestions have been useful and fully considered by the authors.

I strongly recommend the authors to include the uncertainty maps produced for the prediction of the distribution of each mosquito species. Pairing prediction and uncertainty in the same figure is a nice way to assess the fiability of the results (as areas with high uncertainty should be considered unreliable).  

Author Response

Dear reviewer 1, 

Thank you for you rapid reply and further feedback. Initial runs of Maxent were performed by using 75% of occurrence records for model training and 25% for testing. To create the uncertainty maps as requested, we ran Maxent using the k-fold cross validation settings where the number of replicates was set to 10. We have included these niche model predictions and uncertainty maps in the supplementary materials. Interestingly, standard deviation appears to be low between all simulated models. We have decided to keep the focus of our results analysis on our initial run since no sample averages file is produced for the cross-validation average output. However we do note the robustness of cross-validation, and therefore have chosen to provide these results in the supplement and the files to interested readers upon request.